Evaluating active versus passive sources of human brucellosis in Jining City, China

Sun Xihong 1 2
Jiang Wenguo 2
Li Yan 2
Li Xiuchun 3
Zeng Qingyi 4
Du Juan 5 6
Yin Aitian 1 yaitian@sdu.edu.cn
http://orcid.org/0000-0002-2804-0827 Lu Qing-Bin 5 6 qingbinlu@bjmu.edu.cn
1 Centre for Health Management and Policy Research, School of Public Health, Cheeloo College of Medicine, Shandong University, NHC Key Laboratory of Health Economics and Policy Research (Shandong University) , Jinan, Shandong , China
2 Jining Center for Disease Control and Prevention , Jining, Shandong , China
3 Liangshan Center for Disease Control and Prevention , Jining, Shandong , China
4 Yutai Center for Disease Control and Prevention , Jining, Shandong , China
5 Laboratorial Science and Technology, School of Public Health, Peking University , Beijing , China
6 Center for Infectious Disease Research and Policy, Peking University Institute for Global Health , Beijing , China
Odom John Audrey
Electronic publication date: 2021 Jun 22
Publication date: 2021
Volume: 9
Electronic Location ID: e11637
Received 2020 Dec 26; Accepted 2021 May 27
Copyright: © 2021 Sun et al.
Copyright year: 2021
Copyright holder: Sun et al.
License: This is an open access article distributed under the terms of the Creative Commons Attribution License, which permits unrestricted use, distribution, reproduction and adaptation in any medium and for any purpose provided that it is properly attributed. For attribution, the original author(s), title, publication source (PeerJ) and either DOI or URL of the article must be cited.
License URL: https://creativecommons.org/licenses/by/4.0/

Keywords: Human brucellosis, Active and passive sources, Contact history, China

Funding: Shandong Medical and Health Science and Technology Development Project 2019WS095 Peking University Medicine Fund of Fostering Young Scholars’ Scientific & Technological Innovation BMU2021PY005 National Natural Science Foundation of China 82073617 and 81703274 Joint Research Fund for Beijing Natural Science Foundation and Haidian Original Innovation L202007 This work was supported by Shandong Medical and Health Science and Technology Development Project (No. 2019WS095), Peking University Medicine Fund of Fostering Young Scholars’ Scientific & Technological Innovation (BMU2021PY005), National Natural Science Foundation of China (Nos. 82073617 and 81703274) and the Joint Research Fund for Beijing Natural Science Foundation and Haidian Original Innovation (No. L202007). The funders had no role in study design, data collection and analysis, decision to publish, or preparation of the manuscript.

==============================
Human brucellosis (HB) remains a serious public health concern owing to its resurgence across the globe and specifically in China. The timely detection of this disease is the key to its prevention and control. We sought to describe the differences in the demographics of high-risk populations with detected cases of HB contracted from active versus passive sources. We collected data from a large sample population from January to December 2018, in Jining City, China. We recruited patients that were at high-risk for brucellosis from three hospitals and Centers of Disease Control and Prevention (CDCs). These patients were classified into two groups: the active detection group was composed of individuals receiving brucellosis counseling at the CDCs; the passive detection group came from hospitals and high-risk HB groups. We tested a total of 2,247 subjects and 13.3% (299) presented as positive for HB. The positive rates for active and passive detection groups were 20.5% (256/1,249) and 4.3% (43/998), respectively (p < 0.001). The detection rate of confirmed HB cases varied among all groups but was higher in the active detection group than in the passive detection group when controlled for age, sex, ethnicity, education, career, and contact history with sheep or cattle (p < 0.05). Males, farmers, those with four types of contact history with sheep or cattle, and those presenting fever, hyperhidrosis and muscle pain were independent factors associated with confirmed HB cases in multivariate analysis of the active detection group. Active detection is the most common method used to detect brucellosis cases and should be applied to detect HB cases early and avoid misdiagnosis. We need to improve our understanding of brucellosis for high-risk populations. Passive HB detection can be supplemented with active detection when the cognitive changes resulting from brucellosis are low. It is important that healthcare providers understand and emphasis the timely diagnosis of HB.

Introduction

Human brucellosis (HB) is one of the most understudied tropical diseases. It is caused by the genus Brucella and results in large economic losses in the livestock industry and incredible illness in humans who contract the disease (Ariza et al., 2007; Franco et al., 2007; Mableson et al., 2014). The disease is transmitted to humans through close contact with sick animals or the consumption of contaminated raw meat or dairy products (Deng et al., 2019; Dadar, Shahali & Whatmore, 2019). The threat of brucellosis varies among occupational groups and the general population (Chen et al., 2014). Brucellosis is the most prevalent and tenth most prevalent disease among impoverished populations in a study of 76 animal diseases and syndromes, respectively (Molyneux et al., 2011). HB patients present with fever, sweating, fatigue, osteoarticular pain (Zhen et al., 2013; Zheng et al., 2018) and a variety of other complications if not treated in a timely manner (Breivik et al., 2006). Chronic HB results in a wide range of pathological conditions, including spondylitis, endocarditis, and meningoencephalitis, and can affect most organs (Pappas et al., 2005). HB remains one of the most significant public health concerns in the world. The World Health Organization (WHO) estimates that a quarter of cases are unreported with 500,000 registered cases per year. The number of unreported cases with unspecified clinical symptoms is estimated to be ten times higher than this figure (FAO, WOAH & WHO, 2006). HB has been on the rise since 2000 and is the only traditional statutory infectious disease in China where the current incidence level is higher than the historical record. Brucellosis morbidity has increased since the middle- and late-1990s and is one of the top ten total cases of Class A and B national notifiable infectious diseases reported in the mainland of China (Lai et al., 2017; Wang et al., 2018). Shandong was among the top 10 provinces reporting cases of HB but the disease has been reported in all cities in Shandong province since 2005 (Yang et al., 2015). It is crucial to reduce this disease’s morbidity through early detection and treatment.

HB is often misdiagnosed, or the diagnosis is delayed due to the disease’s atypical symptoms (Pappas et al., 2006). Cases that are not diagnosed and treated in a timely manner will become chronic and reoccur, so a timely diagnosis is necessary.

The Jining CDC and county-level CDC agencies monitor HB cases in accordance with the requirements of the Shandong CDC in China. These centers have assisted medical institutions in the diagnosis and treatment of HB for 6 consecutive years since 2013. CDC agencies provide serological testing and individual brucellosis counseling when HB is suspected. Patients presenting clinical symptoms may choose to go to the hospital to be tested and treated. High-risk populations are administered regular tests by the CDC for routine monitoring since they are considered to be passive sources of the disease. We conducted this study to evaluate the effectiveness of HB disease-detection using active sources, as determined by individual brucellosis counseling, and passive sources, based on hospital and CDC reporting.

Materials and Methods

Study population

We enrolled participants from different testing sources who tested for HB from January to December 2018 in Jining City (Fig. 1). We obtained 1,249 participants from the eight CDC centers in Jining City. Participants were tested to determine whether they were active HB cases. These subjects were classified as the active detection group. The other 998 participants were obtained from the three hospitals or CDC staff using passive means, such as a medical diagnosis (n = 480) or staff reporting for HB disease detection (n = 518). These cases were classified as the passive detection group and were obtained from suspected cases that were diagnosed in the hospital or from high-risk rural groups identified through CDC surveillance.

Figure 1 The flowchart of the enrollment of all people tested for brucellosis from different sources during January to December 2018 in Jining City.

Case definition and data collection

Brucellosis cases were subject to a defined brucellosis diagnosis (2012 edition) (National Health Commission of China, 2012). Suspected cases were identified as being any patient with acute or insidious fever, clinical manifestations of HB, and an epidemiological link with infected animals, contaminated food, or contact with a confirmed case. Confirmed cases were defined as having a titer of greater than 1:100 in the test-tube agglutination test (SAT), or a patient sample titer of more than 1:50 with a course of disease lasting more than 1 year. We collected a blood sample of at least 3 mL from each participant using a vacutainer needle. We performed the Rose Bengal plate test (RBPT) and serum agglutination test (SAT) as previously reported (Alton, Jones & Pietz, 1975). All the reagents were configured and provided by the Shandong CDC.

A self-designed questionnaire was used to collect information, including socio-demographic characteristics, animal exposure, and clinical manifestations (Table S1).

Data management and analysis

Data was entered using EpiData 3.1 software. We performed a descriptive analysis of the independent variables for age, sex, nation, education, career, and history of contact. Quantitative variables were expressed as mean with standard deviation (SD) or median with interquartile range (IQR) according to the normality of the variable. Categorical data were analyzed using a chi-square or Fisher exact test. Univariate analysis was done to evaluate the associations between patients’ characteristics and other variables. Covariates associated with p value ≤ 0.2 in the univariate analysis were entered into a multivariate logistic regression model with the stepwise method to obtain the final predictive model of covariates independently associated with HB. We estimated the odds ratio (OR) and 95% confidence interval (95% CI). The area under the curve (AUC) was used to evaluate the prediction accuracy of the model by receiver operating characteristic curve (ROC) analysis. A two-sided p value of less than 0.05 was considered significant. All statistical analyses were done using SPSS software (version 22.0).

Ethics statement

The research protocol was approved by the Peking University Institutional Review Board (IRB00001052-20008). The Research Ethics Committee waived the informed consent requirement because data were legally obtained from the CDC. All the data were analyzed anonymously.

Results

Participant characteristics

The median age of the study participants was 48 (IQR 37-56) years and there were 1,504 (66.9%) males studied (Table 1). The majority (2,174, 96.7%) of the patients were of Han nationality and 1,195 (53.2%) participants had a median education level (with 9 years education). Approximately half of the participants (1,121, 49.9%) were farmers and 1,565 (69.7%) participants had contact history with sheep or cattle. There were significant differences in age, ethnicity, education, career, and contact history with sheep or cattle between the active and passive detection groups and the participants from the hospital and CDC in the passive detection group (all p < 0.05). There were more clinical manifestations, including fever, arthralgia, debilitation, hyperhidrosis and muscle pain, observed in the active detection group than in the passive detection group (p < 0.05).

Table 1 The characteristics of the recruited participants from different sources.

Characteristics	Total (N = 2,247)	Active (n = 1,249)	Passive (n = 998)	p	Passive	
From hospital (n = 480)	From CDC (n = 518)	p	
Age, year, median (IQR)	48 (37–56)	46 (35–56)	48 (39–56)	0.016*	49 (40–60)	48 (38–55)	0.059*	
≤45	959 (42.7)	563 (45.1)	396 (39.7)	<0.001#	181 (37.7)	215 (41.5)	<0.001#	
45~60	900 (40.1)	461 (36.9)	439 (44.0)		183 (38.1)	256 (49.4)		
>60	388 (17.3)	225 (18)	163 (16.3)		116 (24.2)	47 (9.1)		
Sex				0.528#			<0.001#	
Male	1504 (66.9)	843 (67.5)	661 (66.2)		280 (58.3)	381 (73.6)		
Female	743 (33.1)	406 (32.5)	337 (33.8)		200 (41.7)	137 (26.4)		
Ethnicity				0.001#			0.004#	
Han	2174 (96.7)	1195 (95.7)	979 (98.1)		473 (98.5)	506 (97.7)		
Hui	73 (3.2)	54 (4.3)	19 (1.9)		7 (1.5)	12 (2.3)		
Education				0.001#			<0.001#	
Under junior middle school	633 (28.2)	379 (30.3)	254 (25.5)		167 (34.8)	87 (16.8)		
Junior middle school	1195 (53.2)	619 (49.6)	576 (57.7)		201 (41.9)	375 (72.4)		
Above junior middle school	419 (18.7)	251 (20.1)	168 (16.8)		112 (23.3)	56 (10.8)		
Career				<0.001#			<0.001#	
Famer	1121 (49.9)	740 (59.3)	381 (38.2)		221 (46)	160 (30.9)		
Worker	709 (31.6)	267 (21.4)	442 (44.3)		107 (22.3)	335 (64.7)		
Children/Student	54 (2.4)	31 (2.5)	23 (2.3)		23 (4.8)	0 (0)		
Veterinarian	59 (2.6)	36 (2.9)	23 (2.3)		1 (0.2)	22 (4.3)		
Other	304 (13.5)	175 (14)	129 (12.9)		128 (26.7)	1 (0.2)		
Contact history of sheep or cow	1565 (69.7)	637 (63.8)	928 (74.3)	<0.001#	135 (28.1)	502 (96.9)	<0.001#	
Cultivation	1135 (50.5)	507 (50.8)	628 (50.3)	0.806#	105 (21.9)	402 (77.6)	<0.001#	
Slaughter	266 (11.8)	76 (7.6)	190 (15.2)	<0.001#	10 (2.1)	66 (12.7)	<0.001#	
Selling	151 (6.7)	49 (4.9)	102 (8.2)	0.002#	8 (1.7)	41 (7.9)	<0.001#	
Process	69 (3.1)	29 (2.9)	40 (3.2)	0.685#	15 (3.1)	14 (2.7)	0.691#	
Clinical manifestation	1435 (63.9)	957 (76.6)	478 (47.9)	<0.001#	476 (99.2)	2 (0.4)	<0.001†	
Fever	1020 (45.4)	608 (48.7)	412 (41.3)	<0.001#	412 (85.8)	0 (0)	<0.001†	
Arthralgia	623 (27.7)	468 (37.5)	155 (15.5)	<0.001#	155 (32.3)	0 (0)	<0.001†	
Debilitation	465 (20.7)	357 (28.6)	108 (10.8)	<0.001#	106 (22.1)	2 (0.4)	<0.001†	
Hyperhidrosis	340 (15.1)	266 (21.3)	74 (7.4)	<0.001#	74 (15.4)	0 (0)	<0.001†	
Muscle pain	327 (14.6)	259 (20.7)	68 (6.8)	<0.001#	66 (13.8)	2 (0.4)	<0.001†	
Other	135 (6.0)	68 (5.4)	67 (6.7)	0.208#	67 (14.0)	0 (0)	<0.001†	
Notes:

* Compared by Mann–Whitney U test.

# Compared by Chi-square test.

† Compared by Fisher exact test.

IQR, interquartile range.

Detection of HB

Among the 2,247 participants tested for HB, 299 (13.3%, 95% CI [11.9–14.8]) were positive and confirmed as having HB. Positive HB cases in the active and passive detection groups were 20.5% (256/1,249, 95% CI [18.3–22.8]) and 4.3% (43/998, 95% CI [3.1–5.8]), respectively (p < 0.001). The percentage of positive HB tests varied for all participants, although there were more positive cases in the active detection group than in the passive detection group when accounting for age, sex, ethnicity, education, career, and contact history with sheep or cattle (p < 0.05) (Table 2). There was an increase in positive HB tests as age increased and years of education decreased either in the active or passive detection group (p < 0.001 for both). The male, or Hui ethnicity, or farmers participants had higher percentages of positive HB tests in the active detection group, which was similar to the passive detection group results. The percentage of positive HB tests due to contact with sheep or cattle was 16.0% (95% CI [14.3–18.0]), which was greater than the 7.0% (95% CI [5.2–9.2]) among those with no contact. Exposure to animal slaughter showed the highest percentage of positive HB tests (21.1%, 95% CI [16.3–26.5]), followed by exposure through sales (18.5%, 95% CI [12.7–25.7]), processing (17.4%, 95% CI [9.3–28.4]), and cultivation (14.3%, 95% CI [12.3–16.4]). Similar results were observed among the two detection groups.

Table 2 The positive detection rates of human brucellosis infection from different sources.

Characteristics	Total (N = 2,247)	Active (n = 1,249)	Passive (n = 998)	p	Passive	
From hospital (n = 480)	From CDC (n = 518)	p	
All the participants	299 (13.3)	256 (20.5)	43 (4.3)	<0.001	37 (7.7)	6 (1.2)	<0.001	
Age, year								
≤45	100 (10.4)	82 (14.6)	18 (4.6)	<0.001	13 (7.2)	5 (2.3)	0.021	
45~60	124 (13.8)	112 (24.3)	12 (2.7)	<0.001	11 (6)	1 (0.4)	<0.001	
>60	75 (19.3)	62 (27.6)	13 (8.0)	<0.001	13 (11.2)	0 (0)	0.021	
Sex								
Male	218 (14.5)	186 (22.1)	32 (4.8)	<0.001	26 (9.3)	6 (1.6)	<0.001	
Female	81 (10.9)	70 (17.2)	11 (3.3)	<0.001	11 (5.5)	0 (0)	0.004	
Ethnicity								
Han	282 (13.0)	240 (20.1)	42 (4.3)	<0.001	36 (7.6)	6 (1.2)	<0.001	
Hui	17 (23.3)	16 (29.6)	1 (5.3)	0.032	1 (14.3)	0 (0)	0.368	
Education								
Under junior middle school	117 (18.5)	103 (27.2)	14 (5.5)	<0.001	14 (8.4)	0 (0)	0.003	
Junior middle school	148 (12.4)	125 (20.2)	23 (4.0)	<0.001	18 (9.0)	5 (1.3)	<0.001	
Above junior middle school	34 (8.1)	28 (11.2)	6 (3.6)	0.006	5 (4.5)	1 (1.8)	0.665	
Career								
Famer	219 (19.5)	197 (26.6)	22 (5.8)	<0.001	22 (10)	0 (0)	<0.001	
Worker	34 (4.8)	22 (8.2)	12 (2.7)	0.002	6 (5.6)	6 (1.8)	0.045	
Children/Student	8 (14.8)	7 (22.6)	1 (4.4)	0.119	1 (4.4)	0 (0)	1.000	
Veterinarian	6 (10.2)	5 (13.9)	1 (4.4)	0.389	1 (100)	0 (0)	0.043	
Other	32 (10.5)	25 (14.3)	7 (5.4)	0.013	7 (5.5)	0 (0)	1.000	
Contact history of sheep or cow								
Yes	251 (16.0)	216 (23.3)	35 (5.5)	<0.001	29 (21.5)	6 (1.2)	<0.001	
Cultivation	162 (14.3)	133 (21.2)	29 (5.7)	<0.001	23 (21.9)	6 (1.5)	<0.001	
Slaughter	56 (21.1)	54 (28.4)	2 (2.6)	<0.001	2 (20.0)	0 (0)	0.016	
Selling	28 (18.5)	27 (26.5)	1 (2)	<0.001	1 (12.5)	0 (0)	0.163	
Process	12 (17.4)	9 (22.5)	3 (10.3)	0.218	3 (20.0)	0 (0)	0.224	
No	48 (7)	40 (12.5)	8 (2.2)	<0.001	8 (2.3)	0 (0)	1.000	
Clinical manifestation	265 (18.5)	226 (23.6)	39 (8.2)	<0.001	37 (7.8)	2 (100)	0.006	
Fever	197 (19.3)	171 (28.1)	26 (6.3)	<0.001	26 (6.3)	0 (0)	1.000	
Arthralgia	130 (20.9)	104 (22.2)	26 (16.8)	0.148	26 (16.8)	0 (0)	1.000	
Debilitation	93 (20.0)	79 (22.1)	14 (13.0)	0.037	12 (11.3)	2 (100)	0.016	
Hyperhidrosis	102 (30.0)	92 (34.6)	10 (13.5)	<0.001	10 (13.5)	0 (0)	1.000	
Muscle pain	93 (28.4)	83 (32.1)	10 (14.7)	0.004	8 (12.1)	2 (100)	0.020	
Other	8 (5.9)	5 (7.4)	3 (4.5)	0.718	3 (4.5)	0 (0)	1.000	

There was a 7.7% (95% CI [5.5–10.5]) positive HB rate obtained from the hospital among the passive detection group, which was greater than the 1.2% (95% CI [0.4–2.5]) obtained from the CDC (p < 0.001). The former was higher than the latter with variations in age, sex, ethnicity, education, career, and contact history with sheep or cattle but most of the differences were not significant due to the small sample size.

Participants with hyperhidrosis tended to have the highest percentage of positive HB tests (30.0%), followed by muscle pain (28.4%), arthralgia (20.9%), debilitation (20.0%), and fever (19.3%). These results were mirrored in the active detection group. Positive HB tests in the passive detection group, including arthralgia (16.8%), muscle pain (14.7%), hyperhidrosis (13.5%), debilitation (13.0%) and fever (6.3%), were lower than in the active detection group.

Factors associated with the risk of HB

Multivariate analysis determined that the male sex, farmer, four types of contact history with sheep or cattle, and the presentation of fever, hyperhidrosis, and muscle pain were independently-associated factors with confirmed HB. The active detection group’s population source with an AUC of 0.802 (0.777–0.827) (all p < 0.05) is shown in Table 3. Selling lamb and beef presented the highest risk of HB (OR = 3.54, 95% CI [2.06–6.08]) in the four types of contact history with sheep or cattle. HB risk factors in the active detection group were the male sex, farmer, two types of contact history with sheep or cattle (slaughter and selling), and the presentation of fever, arthralgia and debilitation with an AUC of 0.760 (95% CI [0.732–0.788]) (Table 3). HB risk within the passive detection group was significantly associated with younger age, cultivation of sheep or cattle, and arthralgia with an AUC of 0.673 (95% CI [0.639–0.706]) (all p < 0.05). The cultivation of sheep or cattle and hyperhidrosis were associated with the risk of HB infection in the passive detection group (Table S2).

Table 3 The independent factors related to the positive detection of human brucellosis in the active and passive detection groups.

Characteristics	Total	Active detection group	Passive detection group	
OR (95% CI)	p	OR (95% CI)	p	OR (95% CI)	p	
Age, year							
≤45	Reference		Reference		Reference		
45~60	1.2 [0.87–1.65]	0.262	1.40 [0.99–2.00]	0.060	0.42 [0.18–0.98]	0.044	
>60	1.34 [0.89–2.02]	0.167	1.40 [0.88–2.23]	0.152	0.80 [0.30–2.14]	0.657	
Sex							
Male	1.48 [1.09–2.01]	0.011	1.41 [1.01–1.98]	0.043	2.19 [0.94–5.13]	0.070	
Female	Reference		Reference		Reference		
Ethnicity							
Han	0.71 [0.38–1.33]	0.281	0.71 [0.37–1.38]	0.314	0.66 [0.06–7.18]	0.729	
Hui	Reference		Reference		Reference		
Education							
Under junior middle school	Reference		Reference		Reference		
Junior middle school	0.93 [0.67–1.29]	0.658	0.85 [0.59–1.21]	0.367	1.03 [0.42–2.55]	0.944	
Above junior middle school	0.72 [0.44–1.17]	0.185	0.64 [0.37–1.11]	0.113	1.06 [0.31–3.61]	0.927	
Career							
Worker	Reference		Reference		Reference		
Farmer	2.39 [1.54–3.69]	<0.001	2.66 [1.58–4.50]	<0.001	1.39 [0.52–3.68]	0.511	
Children/Student	1.51 [0.6–3.85]	0.383	1.74 [0.62–4.93]	0.294	0.74 [0.04–12.93]	0.835	
Veterinarian	1.97 [0.75–5.15]	0.169	2.38 [0.80–7.05]	0.118	2.01 [0.24–17.08]	0.523	
Other	1.72 [0.96–3.08]	0.067	1.59 [0.80–3.15]	0.182	2.57 [0.75–8.80]	0.133	
Contact history of sheep or cow							
Cultivation	1.78 [1.28–2.47]	0.001	1.43 [0.99–2.07]	0.054	7.44 [2.87–19.25]	<0.001	
Slaughter	1.85 [1.23–2.79]	0.003	1.98 [1.26–3.09]	0.003	0.95 [0.17–5.23]	0.951	
Selling	2.58 [1.49–4.48]	0.001	2.37 [1.32–4.22]	0.004	3.71 [0.36–38.8]	0.273	
Process	2.37 [1.1–5.1]	0.028	1.92 [0.78–4.73]	0.157	4.60 [0.74–28.38]	0.100	
Clinical manifestation							
Fever	2.04 [1.51–2.76]	<0.001	1.91 [1.38–2.65]	<0.001	1.67 [0.700–4.00]	0.246	
Arthralgia	1.28 [0.96–1.69]	0.087	0.9 [0.66–1.23]	0.509	10.35 [4.63–23.15]	<0.001	
Debilitation	0.84 [0.61–1.15]	0.281	0.76 [0.54–1.06]	0.105	1.97 [0.78–4.95]	0.152	
Hyperhidrosis	1.81 [1.3–2.52]	<0.001	1.91 [1.34–2.73]	<0.001	1.47 [0.50–4.35]	0.481	
Muscle pain	1.69 [1.23–2.33]	0.001	1.55 [1.10–2.18]	0.013	1.80 [0.61–5.34]	0.291	
Other	0.41 [0.19–0.87]	0.02	0.29 [0.11–0.75]	0.011	1.37 [0.35–5.30]	0.651	
Population source							
Active detection group	3.45 [2.41–4.96]	<0.001					
Passive detection group	Reference						
Note:

OR, odds ratio; CI, confidence interval. Other clinical manifestation includes headache, anemia, diarrhea, thrombocytopenia, rash, loss of consciousness, vomit and edema.

Discussion

HB is a major public health crisis in China. Although its true incidence remains largely unknown, it is conservatively estimated to vary from less than 0.03 to more than 160 individuals per 100,000 (Pappas et al., 2006; Lai et al., 2017; Wang et al., 2020). It is important to have early detection and treatment of suspected cases. We investigated whether the detection rate of HB was higher in the active detection group than in the passive detection group. The active detection group was typically aware of the CDC’s role as an institution for the detection of brucellosis, suggesting that the active detection method was better able to detect HB based on greater HB awareness for this high-risk population.

A meta-analysis reported low levels of awareness and knowledge of HB among high-risk groups in Asia and Africa (Zhang et al., 2019), which was thought to be affected by many factors and is an obstacle for public health. Raising awareness of brucellosis is of great significance for controlling HB. HB prevention was publicized to high-risk populations in the study area to assist in early detection and improve the awareness of HB. Those who were knowledgeable about HB were more likely to use a detection service when clinical symptoms that may be related to HB first appeared. This might be due to the fact that the positive rate was high than among the high-risk population in the active detection group. Thus, active detection is effective for the early detection of HB and publicity and education about HB may be effective in the affected areas.

We found a much lower HB detection rate in the passive detection group, including two different participant sources from hospitals and the local CDC. However, the cause of the low detection rate of HB between participants from these two sites may be different. The participants from hospitals had similar clinical symptoms but had less contact history with sheep or cattle compared to the participants from the local CDC. This result revealed a low detection rate and may cause a high rate of HB misdiagnosis based solely on clinical manifestation. Participants from the local CDC, however, had a higher frequency of contact history with sheep or cattle but seldom presented with clinical manifestations of HB compared to the participants from hospitals. These results revealed a low detection rate of HB based on contact history with sheep or cattle. Active detection methods avoid the disadvantages of the two passive detection methods and should be utilized on a larger scale. Active detection is also an important way to detect cases by doctors from hospitals, considering the 7.7% detection rate in the hospital. Primary healthcare workers should be aware of the symptoms of acute and chronic brucellosis when treating patients (Yumuk & O’Callaghan, 2012). The passive detection of HB by the CDC was more successful in a specific population.

Our findings showed that sex, career, contact history with sheep or cattle, and the presentation of specific symptoms (fever, hyperhidrosis and muscle pain) were HB risk factors. This is in agreement with findings from other studies (Cao et al., 2017; Cash-Goldwasser et al., 2018; Alkahtani et al., 2020).

Males had the highest disease incidence which may be a reflection of their occupational exposure to livestock in a pastoral economy where females typically have less exposure to livestock (Makita et al., 2008). Farmers, who were typically older, developed HB more frequently. It is common for the elderly in rural areas in northern China to raise cattle and sheep through family farming. Our results indicated that efforts, such as increased publicity, education for farmers in contact with cattle, and raising awareness of disease prevention through personal protection, must be taken for the high-risk population.

Brucellosis is one of the most common zoonotic infections and contact with sheep or cattle is the most important HB risk factor. Of the four types of contact history with sheep or cattle, the sale of livestock showed the highest OR of HB, which may be related to longer exposure time to lamb or beef. Slaughtering and processing also showed higher ORs than cultivation. The HB detection rates in the slaughter, selling, and processing population will be lower when the detection of Brucella before slaughter is improved.

We analyzed the characteristics of a high-risk population to improve the use of limited healthcare resources and achieve better prevention and control of HB as diagnosing Brucella in the laboratory is not in wide use. It is necessary to improve the monitoring of those with clear occupational exposure history, especially those with clinical symptoms such as hyperhidrosis for better passive detection. Patients can improve their awareness of HB through health education and then proceed to active detection methods for better early diagnosis. Meeting the medical needs of active cases is the priority for treatment of HB. Behavioral intervention should be strengthened to improve treatment compliance and prevent chronic cases and recurrence of the disease. The two detection methods have different focuses in the prevention and control of HB. Passive detection mode sensitivity is low, but it is an important supplement when the cognitive impairment of brucellosis is still low. The current brucellosis surveillance pattern in China includes a passive surveillance system based on the epidemic and an active surveillance system in key areas that include monitoring stations (Dong, Jiang & Wang, 2019). It is necessary to choose the appropriate detection strategy of HB to improve the accuracy of monitoring data.

Our study provides valuable information about HB detection in Jining City, however, there were some limitations. First, our study did not account for more high-risk population for HB in the community when the population did not visit the health facilities including hospitals and CDCs. Second, we did not investigate the subjects’ awareness and knowledge of HB, which may be related to the risk of contracting HB. Third, we did not culture for Brucella spp. to give a gold standard result, which may have a certain bias on our reported prevalence.

We determined three measures for the control and prevention of HB in this region. First, the public should be provided with more health education to improve the level of self-protection and the awareness of timely diagnosis and treatment. Second, hospital healthcare providers should be more aware of HB for a timely and accurate diagnosis. Third, the CDC, as the prevention and control department of HB, should improve the breadth and depth of brucellosis detection and screening to accurately diagnosis HB cases.

Conclusion

HB is a serious public-health problem owing to its resurgence in China and across the globe. We clearly demonstrated that the active detection model more accurately detects cases of HB than the passive detection model. We also revealed that those who had contact with sheep or cattle had the highest risk of contracting HB. These results may help policy makers develop appropriate prevention and control strategies.

Supplemental Information

Supplemental Information 1 Self-designed Questionnaire.

Click here for additional data file.

Supplemental Information 2 The associated factors related to the positive detection of human brucellosis in passive surveillance group from hospital.

Click here for additional data file.

Supplemental Information 3 Database.

Click here for additional data file.

Supplemental Information 4 The self-made questionnaire for collecting information.

Click here for additional data file.

Additional Information and Declarations

Competing Interests

Author Contributions

Human Ethics

Data Availability

The authors declare that they have no competing interests.

Xihong Sun conceived and designed the experiments, performed the experiments, analyzed the data, prepared figures and/or tables, authored or reviewed drafts of the paper, and approved the final draft.

Wenguo Jiang performed the experiments, prepared figures and/or tables, authored or reviewed drafts of the paper, and approved the final draft.

Yan Li performed the experiments, authored or reviewed drafts of the paper, and approved the final draft.

Xiuchun Li performed the experiments, authored or reviewed drafts of the paper, and approved the final draft.

Qingyi Zeng performed the experiments, authored or reviewed drafts of the paper, and approved the final draft.

Juan Du performed the experiments, analyzed the data, prepared figures and/or tables, authored or reviewed drafts of the paper, and approved the final draft.

Aitian Yin conceived and designed the experiments, prepared figures and/or tables, authored or reviewed drafts of the paper, and approved the final draft.

Qing-Bin Lu conceived and designed the experiments, analyzed the data, prepared figures and/or tables, authored or reviewed drafts of the paper, and approved the final draft.

The following information was supplied relating to ethical approvals (i.e., approving body and any reference numbers):

The research protocol was approved by Peking University Institutional Review Board (IRB00001052-20008).

The following information was supplied regarding data availability:

The raw measurements are available in the Supplemental File.

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
