# Peer review of "Evaluating active versus passive sources of human brucellosis in Jining City, China"

_PeerJ, doi:10.7717/peerj.11637_

## Round 0.1 · original submission · Major Revisions

Please address the key concerns raised by reviewer 2 regarding the rate of detection. In addition, careful copyediting and grammar review are required to improve readability.

Reviewer 1 ·

Basic reporting

English needs to be improved.

Experimental design

no comment.

Validity of the findings

no comment.

Additional comments

This is an interesting paper that provides useful novel information on performance of two detection modes for human brucellosis and the factors associated with human brucellosis in Jining City, China and got the conclusion that active detection mode is better for early diagnosis of HB and therefore proposed three measures including strengthening public education in high-risk area and improving diagnosis in both hospitals and CDCs. The findings are interesting and of significance to prevention and control of HB. Thus it can be considered for publication in this journal after some revision.
Some general comments:
1.Although the language is generally readable, there are many mistakes in spelling and grammar and wrong use. Thus proofreading/editing by a native English speaker is recommended. Some examples are given as follows:
L70 “mortality” should be “morbidity”;
L86-87: “due to the positive cases were not initiative”
2. Please check and ensure the citation of refs is correct. For example Line 91, text mentioned 2018 edition, but the ref 17 is actually 2012 edition.
3. “the positive detection rate” isn’t a typical epidemiological term, it is better to be replaced by percentage of test positive or test prevalence.
4. The test prevalence (or the positive detection rate used in this manuscript) should be displayed with 95% CI.
5. In conclusion, the proposed three measures are not conclusions, thus should be moved into the Discussion section.

Some specific comments:
Abstract
1. Line 27, you should use the term of “high-risk” throughout the manuscript, do not add space inside the phrase.
2. Line 28-29, the sentence of “Data were analyzed that experienced the greatest number of cases during the period under observation” is difficult to understand. Please rewrite this sentence.
3. Line 35, 37, regarding P values, the should be in italics and small letter.

Introduction
4. Line 54-55, here should put “contaminated ” before “raw meat and dairy products”.
5. Line 70, “mortality” should be replaced by “morbidity” because brucellosis rarely results in death in humans.

Materials and Methods
6. Line 86-87, “due to the positive cases ” is suggested to revise “since these cases” .
7. Line 91, the reference biography for the diagnosis criteria for brucellosis in human should be updated to 2018 edition.
8. Line 92-96, please use the correct ref about the 2018 edition.
9. Line 99-103, it is suggested to integrate clinical signs and laboratory procedures into the case definition.
10. Line 110-110, Odds ratio (OR) should be added into the section of data analysis.
11. Line 112, regarding multivariate logistic regression model, please describe how to evaluate the prediction accuracy of the model. Besides, it is suggested to include population source as a covariate to determine whether the active source is more likely to detect HB infection than the passive source.

Results
12. Line 131, clinical manifestations should be written in the plural.
13. Line 132, P values should be in italics.
14. Line 139, with the lower levels of education
15. Line 157-168, the section of factors associated with the risk of HB could be improved with more details after reanalysis.

Discussion
16. Line 172-173, please refer to latest publications in China.
17. Line 194-196, “The participants from hospitals had more similar clinical symptoms and signs but with less contact history of sheep or cow, which may reveal a higher misdiagnosis rate of HB based on the clinical manifestation of HB.” The points are confusing and the sentence has grammar mistakes.
18. Line 197-198, The points are confusing and the sentence has grammar mistakes.
19. Line 220, the first letter of Brucella should be capitalized, and the word should be italicized.
20. Line 223-225, “For passive detection, strengthen the monitoring on the population with a clear history of occupational exposure, especially the co-presenting of hyperhidrosis.” – this sentence is grammatically flawed.
21. Line 241-242, “Third, we did not perform the culture of the Brucellosis to give a gold standard result, which may have a certain bias on the detection rate.” –“Brucellosis” should be replaced with “Brucella spp.”; “detection rate” should be “prevalence”.

Acknowledgment
22. Line 259-260, the font size is not consistent.

Table 1
23. The statistical analysis methods on different types of variables should be described in the footnote.

Figure 1
24. The first letter of Brucella should be capitalized, and the word should be italicized.

Reviewer 2 ·

Basic reporting

no comment

Experimental design

The grouping method is not scientific

Validity of the findings

I don't think the results are consistent with reality.

Additional comments

There were more clinical symptoms in the active detection group than in the passive detection group, and the positive rate of brucellosis was higher than that in the passive detection group. I think these are not consistent with reality. In addition, the method section states that the diagnostic criteria are 2018 (Line 91), while the reference is 2012?

---

## Round 0.2 · accepted · Accept

You have done a nice job responding to the reviewers' queries, and the overall manuscript reads much more cleanly.